# Hemiacetalmeroterpenoids A–C and Astellolide Q with Antimicrobial Activity from the Marine-Derived Fungus *Penicillium* sp. N-5

**DOI:** 10.3390/md20080514

**Published:** 2022-08-13

**Authors:** Tao Chen, Wencong Yang, Taobo Li, Yihao Yin, Yufeng Liu, Bo Wang, Zhigang She

**Affiliations:** School of Chemistry, Sun Yat-sen University, Guangzhou 510275, China

**Keywords:** andrastin-type meroterpenoids, drimane sesquiterpenoid, marine-derived fungus, antimicrobials activities

## Abstract

Four new compounds including three andrastin-type meroterpenoids hemiacetalmeroterpenoids A-C (**1**–**3**), and a drimane sesquiterpenoid astellolide Q (**15**), together with eleven known compounds (**4**–**14**) were isolated from the cultures of the marine-derived fungus *Penicillium* sp. N-5, while compound **14** was first isolated from a natural source. The structures of the new compounds were determined by analysis of detailed spectroscopic data, and the absolute configurations were further decided by a comparison of the experimental and calculated ECD spectra. Hemiacetalmeroterpenoid A (**1**) possesses a unique and highly congested 6,6,6,6,5,5-hexa-cyclic skeleton. Moreover, the absolute configuration of compound **14** was also reported for the first time. Compounds **1**, **5** and **10** exhibited significant antimicrobial activities against *Penicillium italicum* and *Colletrichum gloeosporioides* with MIC values ranging from 1.56 to 6.25 μg/mL.

## 1. Introduction

Andrastins are meroterpenoids characterized by a 6,6,6,5-tetra-carbocyclic skeleton. They are biogenetically derived from 3,5-dimethylorsellinic acid (DMOA) and farnesyl diphosphate (FPP), synthesized via a mixed polyketide-terpenoid pathway, and usually possess a keto-enol tautomerism at the cyclopentane ring [1,2,3,4]. To date, over 40 andrastins have been reported with multiple potential biological activities, including cytotoxic [5], anti-inflammatory [6], antiproliferative [7] and antimicrobial activity [4]. The complex structures and potential biological activities of andrastins have attracted much attention in recent years [8,9,10].

Marine fungus is known to be a natural source of structurally diverse and biologically active metabolites for drug discovery [11,12,13,14,15,16]. Recently, a series of novel bioactive natural products from marine fungi were reported by our group [17,18,19,20,21,22]. In our ongoing search for new bioactive secondary metabolites from marine fungi, the fungus *Penicillium* sp. N-5, isolated from the rhizosphere soil of mangrove plant *Avicennia marina*, led to the isolation of four new compounds, hemiacetalmeroterpenoids A–C (**1**–**3**) and astellolide Q (**15**). Especially, hemiacetalmeroterpenoid A (**1**) was a new andrastin-type meroterpenoid containing a unique 6,6,6,6,5,5-hexa-cyclic skeleton. Meanwhile, eleven known compounds, including 3-deacetyl-citreohybridonol (**4**) [23] citreohybridone A (**5**) [24], 3,5-dimethylorsellinic acid-based meroterpenoid 2 (**6**) [5], andrastins A–C (**7**, **10**, **13**) [25], andrastone C (**8**) [26], penimeroterpenoid A (**9**) [4], 23-deoxocitreohybridonol (**11**) [1], 6α-hydroxyandrastin B (**12**) [1], and compound V (**14**) [27] were also obtained from the fungus N-5 (Figure 1). All the isolated compounds were investigated for their antimicrobial activity against two phytopathogenic fungi and four bacterial strains. Herein, we report the isolation, structural characterization and antibacterial activity of these compounds.

## 2. Results

### 2.1. Structure Identification

Hemiacetalmeroterpenoid A (**1**) was obtained as a white powder. It molecular formula was assigned as C_26_H_34_O_7_ according to HRESIMS analysis at *m*/*z* 459.23709 [M+H]^+^ (calcd. 459.23773), indicating ten degrees of unsaturation. In the ^1^H NMR spectrum, the signal for one olefinic proton (*δ*_H_ 5.63), one methoxyl (*δ*_H_ 3.60), one methine (*δ*_H_ 1.47), four methylenes (*δ*_H_ 1.42, 1.62, 1.85, 1.91, 1.94, 2.11, 2.33 and 2.61) and six methyls (*δ*_H_ 1.03, 1.07, 1.19, 1.23, 1.33 and 1.49). The ^13^C NMR data exhibited 26 carbon resonances, including two olefinic carbons for one double bond (*δ*_C_ 127.2, 150.1), three carbonyl carbons for two ketone (*δ*_C_ 203.7 and 203.8), and one ester carbonyl (*δ*_C_ 169.3), one methine (*δ*_C_ 49.1), five methylenes (one oxygenated), seven methyls (one oxygenated), eight quaternary carbons (one highly oxygenated: *δ*_C_ 99.8) (Table 1). The NMR data established a nucleus of meroterpenoid characterized by an andrastin scaffold, structurally similar to the citreohybriddione C (Appendix A) [28]. Analysis of the ^1^H-^1^H COSY data led to the identification of two isolated spin-systems of C-1/C-2 and C-5/C-6/C-7. The HMBC from H_2_-1 to C-3, C-10, from H_1_-5 to C-10, and from H_3_-22 to C-3, C-4, C-5, C-23, ring A was formed. The HMBC correlations from H_3_-24 to C-7, C-8, C-9, from H_1_-11 to C-8, C-10 and from H_2_-21 to C-5, C-10 completed ring B. Then, the HMBC cross-peaks from H_2_-1, H_2_-6 to C-10 and from H_3_-23, H_2_-7 to C-5 indicated ring A and ring B were fused at C-5 and C-10. Then, HMBC correlations from H_3_-24 to C-14, C-15, from H_3_-20 to C-11, C-12, C-13, C-17, from H_3_-19 to C-12, C-13, C-14, C-17 and from H_3_-18 to C-15, C-16, C-17, ring C and ring D were constituted, and they were blended at C-13 and C-14. The HMBC correlations from H_2_-6, H_1_-11 to C-8, from H_1_-11 to C-10, suggested ring B and ring C were tightly connected. In addition, the HMBC correlation from H_3_-26 to C-25 implied the presence of a methyl carboxylate. A weak HMBC correlation from H_3_-26 to C-14 located the methyl carboxylate at C-14. Except one double bond, three carbonyls and four rings, ten degrees of unsaturation indicated that two new rings were required. According to the HMBC correlation from H_2_-21 to C-3 (*δ*_C_ 99.8), a 6-membered ring was confirmed between C-1, C-2, C-3, C-10, and C-21. Finally, another new 5-membered ring was formed by intramolecular dehydration of hydroxyl groups at C-12 and C-16. Thus, the planar structure of **1** was established as shown in Figure 2.

The relative configuration of compound **1** was defined by the NOESY correlations. The correlations of H_2_-21 with H_3_-23, H_3_-20 with H_3_-19, H_3_-24, and H_3_-18 with H_3_-19, H_3_-26 were observed in the NOESY spectrum, which means H_3_-18, H_3_-19, H_3_-20, H_2_-21, H_3_-23, H_3_-24 and H_3_-26 were on the same side. The NOESY correlations of H_1_-5 with H_3_-22 suggested that H_1_-5 and H_3_-22 were in the opposite face (Figure 3). The absolute configuration of **1** was determined by comparing the calculated ECD spectra generated by the time-dependent density functional theory (TDDFT) for two enantiomers 3*R*, 5*S*, 8*S*, 10*S*, 12*R*, 13*S*, 14*R*, 16*R*-**1a** and 3*S*, 5*R*, 8*R*, 10*R*, 12*S*, 13*R*, 14*S*, 16*S*-**1b** with the experimental one. Finally, the experimental ECD spectrum of **1** was nearly identical to the calculated ECD spectrum for **1a** (Figure 4), clearly suggesting the 3*R*, 5*S*, 8*S*, 10*S*, 12*R*, 13*S*, 14*R*, 16*R* absolute configuration for **1**.

Hemiacetalmeroterpenoid B (**2**) was isolated as a white powder and had a molecular formula of C_26_H_36_O_6_, determined by HRESIMS data *m*/*z* 445.25772 [M+H]^+^ (calcd. 445.25847) with nine degrees of unsaturation. The ^1^H NMR spectrum of **2** displayed the signal for one olefinic proton (*δ*_H_ 5.42), one methoxyl (*δ*_H_ 3.56), two methines (*δ*_H_ 1.33 and 1.89), four methylenes (*δ*_H_ 1.12, 1.33, 1.55, 1.75, 2.07, 2.12, 2.19 and 2.76) and six methyls (*δ*_H_ 1.01, 1.04, 1.18, 1.19, 1.57 and 1.82). The ^13^C NMR data revealed 26 carbon resonances, involving four olefinic carbons for two double bonds (*δ*_C_ 113.5, 124.2, 137.7, 190.7), two carbonyl carbons for one ketone (*δ*_C_ 201.4), one ester carbonyl (*δ*_C_ 172.6) (Table 1). According to 1D NMR and 2D NMR data, the planar structure of **2** was similar to the co-isolated andrastin B (**13**). The obvious difference is that the acetyl group at the C-3 position of compound **2** disappears. Meanwhile, the HMBC from H_2_-21 to C-3 (*δ*_C_ 99.5) also indicated that a new 6-membered ring was formed between C-1, C-2, C-3, C-10 and C-21 (Figure 2).

The NOESY spectrum indicated that H_1_-5, H_1_-9 and H_3_-22 were on the same side based on the correlations of H_1_-5 with H_1_-9 and H_3_-22. On the contrary, it was suggested that H_3_-19, H_3_-21, H_3_-23, H_3_-24, and H_3_-26 were on the other side based on the NOESY correlations of H_2_-21 with H_3_-23 and H_3_-24, along with H_3_-19 with H_3_-24 and H_3_-26 (Figure 3). Thus, the relative configuration of **2** was determined to be 3*R*, 5*S*, 8*S*, 9*R*, 10*S*, 13*R* and 14*R*. The absolute configuration of the stereogenic centers in **2** was assigned as 3*R*, 5*S*, 8*S*, 9*R*, 10*S*, 13*R* and 14*R* by comparing its experimental ECD spectrum with that of the calculated model molecule (Figure 4).

Hemiacetalmeroterpenoid C (**3**) was also purified as a white powder. The molecular formula was specified as C_28_H_38_O_7_ (ten degrees of unsaturation) by HRESIMS (*m*/*z* 509.25015 [M+Na]^+^), which is 42 mass units higher than that of **2** (Appendix A). Analysis of its NMR data (Table 1) revealed the presence of the same partial structure as that found in compound **2**. The only difference was **3** has an additional acetyl fragment. Finally, a weak HMBC correlation from Ac-CH_3_ to C-15 suggested that the acetyl fragment was attached to C-15 (Figure 2).

Because compound **3** has the same chiral center as **2**, the NOESY correlation and experimental ECD spectrum of compound **3** were in agreement with those of **2** (Figure 3 and Figure 4). Thus, the absolute configuration of **3** was identified as 3*R*, 5*S*, 8*S*, 9*R*, 10*S*, 13*R* and 14*R*.

Compound **14** was obtained as a yellow powder. Analysis of its ^1^H NMR and ^13^C NMR data showed that the planar structure of **14** was the same as compound V, which was the product of the alkaline hydrolysis of parasiticolide A [27]. However, the absolute configuration of compound V was ambiguous.

The relative configuration of **14** was also defined by the NOESY correlation. The correlations of H_3_-14 with H_1_-5 and H_1_-6, and H_2_-13 with H_2_-15 were found in the NOESY spectrum, which means H_1_-5, H_1_-6, and H_3_-4 were on the same side, and H_2_-13 and H_2_-15 were on the opposite face (Figure 3). Thus, the absolute configuration of the stereogenic centers in **14** was assigned as 4*R*, 5*R*, 6*S*, 10*S* by comparing its experimental ECD spectrum with that of the calculated model molecule (Figure 4). Finally, compound **14** was named as astellolide J.

Astellolide Q (**15**) was also acquired as a yellow powder. It molecular formula was determined as C_17_H_24_O_6_ according to HRESIMS analysis at *m*/*z* 347.14578 [M+Na]^+^ (calcd. 347.14651), indicating six degrees of unsaturation. The ^1^H NMR of **15** showed two methyls (*δ*_H_ 1.15 and 2.08), four methylenes (*δ*_H_ 1.16, 1.45, 1.54, 1.78, 1.91, 2.05, 2.34 and 2.50), one methines (*δ*_H_ 1.74) one hydroxymethine (*δ*_H_ 4.55) and three hydroxy-methylenes (*δ*_H_ 3.34, 3.92, 4.09, 4.33, 4.84 and 5.04). In addition, according to the HSQC data, the ^13^C NMR data showed the presence of 17 carbon signals, including two ester carbonyl carbons (*δ*_C_ 173.1, 177.0) and two olefinic carbons (*δ*_C_ 124.0, 169.0), one methyl, seven methylenes (three oxygenated), two methines (one oxygenated), two aliphatic quaternary carbons (Table 2). Analysis of its ^1^H NMR and ^13^C NMR data in association with the 2D NMR data established a nucleus of drimane sesquiterpenoid characterized by an astellolide scaffold, structurally similar to the co-isolated compound **14** (Appendix A). It can be clearly observed that compound **15** has an additional acetyl fragment. Furthermore, the HMBC from H_2_-13 to Ac-OCO indicated that the acetyl fragment was linked to C-13 (Figure 3).

Finally, the NOESY correlation and experimental ECD spectrum of compound **15** were identical to those of **14** (Figure 3 and Figure 4). Thus, the absolute configuration of **15** was also assigned as 4*R*, 5*R*, 6*S*, 10*S*.

### 2.2. Antimicrobial Assay

Compounds **1**–**15** were investigated for their antimicrobial activities against two phytopathogenic fungi and four bacterial strains. As shown in Table 3, andrastin-type meroterpenoids have better antimicrobial activities against phytopathogenic fungus than against bacteria. Most of all the tested compounds (9 compounds out of total 15 compounds) displayed potent antimicrobial activities (MIC < 50 μg/mL). Among them, compounds 1, 5 and 10 exhibited remarkable antimicrobial activities against *Penicillium italicum* and *Colletrichum gloeosporioides* with MIC values of 6.25, 1.56, 6.25 and 6.25, 3.13, 6.25 μg/mL. Moreover, compound 1 showed inhibitory activities against *Bacillus subtilis* under concentration of 6.25 μg/mL. Compound 10 also displayed significant antimicrobial activity against *Salmonella typhimurium* with an MIC value of 3.13 μg/mL. Notably, compound 5 revealed potential antimicrobial activity against all the strains, the MIC values were lower than 25 μg/mL.

As for the study of the structure–activity relationship (SAR), it was found that the degree of oxidation at C-21 had different effects on the activities of the compounds. The compound with methyl (**10**) at C-21 has significantly antimicrobial activity, followed by the aldehyde group (**7**), and hydroxymethyl (**13**) was the weakest. In-depth analysis showed that apart from the degree of oxidation at C-21, keto-enol tautomerism at the cyclopentane ring also had obvious influences on the antimicrobial activities of compounds. Compared to compounds **7** and **13** (enol form), compounds **8** and **9** (keto form) showed no activities against all strains (Table 3).

## 3. Experimental Methods

### 3.1. General Experimental Procedures

The NMR were tested on a Bruker Avance 600 MHz spectrometer (Karlsruhe, Germany) at room temperature. Optical rotations data were recorded on an MCP300 (Anton Paar, Shanghai, China). UV were tested using a Shimadzu UV-2600 spectrophotometer (Shimadzu, Kyoto, Japan). IR spectra were recorded on IR Affinity-1 spectrometer (Shimadzu, Kyoto, Japan). HR-ESI-MS spectra were tested on a ThermoFisher LTQ-Orbitrap-LC-MS spectrometer (Palo Alto, CA, USA). LC-MS/MS data was performed on a Q-TOF manufactured by Waters and a Waters Acquity UPLC BEH C18 column (1.7 µm, 2.1 × 100 mm). Recoated silica gel plates (Qingdao Huang Hai Chemical Group Co., Qingdao, China, G60, F-254), Column chromatography (CC) and Sephadex LH-20 (Amersham Pharmacia, Stockholm, Sweden) were used to purify the compounds. 

### 3.2. Fungal Material

Fungus N-5 was isolated from the rhizosphere soil of mangrove plant *Avicennia marina* (collected in October 2021 from Nansha Mangrove National Nature Reserve in Guangdong Province, China). It was identified as *Penicillum* sp. by the ITS region (deposited in GenBank, accession no ON926808), and fungus N-5 was deposited at Sun Yat-sen University, China.

### 3.3. Fermentation

The fungus *Penicillum* sp. N-5 was cultured in one hundred 1000 mL Erlenmeyer flasks at 25 °C for 30 days; these contained autoclaved rice solid-substrate medium composed of 50 g rice and 50 mL 3‰ saline water. 

### 3.4. Extraction and Purification

After incubation, the mycelia and solid rice medium were extracted four times with EtOAc, and 75 g of residue was obtained. Next, the residue was separated by a gradient of petroleum ether/EtOAc from 9:1 to 0:10 (*v*/*v*) on silica gel CC and divided into six fractions (Fr.1–Fr.6). Fr. 2 (10 g) was separated to Sephadex LH-20 (methanol) to yield three sub-fractions (SFrs. 2.1–2.3). SFrs.2.3 (1.2 g) was applied to silica gel CC (DCM/MeOH *v*/*v*, 100:1) and further purified by reversed-phase (RP) high performance liquid chromatography (HPLC; 90–10% MeCN/H_2_O for 25 min) to obtain compounds **1** (5 mg) and **3** (7 mg). Fr. 3 (16 g) was also separated to Sephadex LH-20 (methanol) to yield four sub-fractions (SFrs. 3.1–3.4). SFrs.3.1 (1.6 g) was separated to silica gel CC (DCM/MeOH *v*/*v*, 80:1) and further purified by reversed-phase (RP) high performance liquid chromatography (HPLC; 75–25% MeCN/H_2_O for 22 min) to yield compounds **2** (11 mg) and **15** (6 mg). 

Hemiacetalmeroterpenoid A (**1**): white powder, m.p. 122.8–124.1 °C; $\;{\left[\alpha \right]}_{D}^{25}$-52 (*c* 0.02, MeOH), UV (MeOH) λ_max_ (log *ε*): 206 (2.52) (Appendix A); ECD (MeOH) λ_max_ (∆*ε*): 240 (+5.47), 301 (−1.14), 362 (+0.61); ^1^H (600 MHz, CD_3_OD) and ^13^C NMR (150 MHz, CD_3_OD) data, see Table 1; HR-ESI-MS: *m*/*z* 459.23709 [M+H]^+^ (calcd. for C_26_H_35_O_7_, 459.23773).

Hemiacetalmeroterpenoid B (**2**): white powder, m.p. 106.9–108.4 °C; $\;{\left[\alpha \right]}_{D}^{25}$ −56 (*c* 0.02, MeOH), UV (MeOH) λ_max_ (log *ε*): 205 (2.83), 238 (1.36) (Appendix A); ECD (MeOH) λ_max_ (∆*ε*): 206 (−11.92), 248 (+3.38), 311 (−1.19); ^1^H (600 MHz, CD_3_OD) and ^13^C NMR (150 MHz, CD_3_OD) data, see Table 1; HR-ESI-MS: *m*/*z* 445.25772 [M+H]^+^ (calcd. for C_26_H_37_O_6_, 445.25847).

Hemiacetalmeroterpenoid C (**3**): white powder, m.p. 100.8–102.6 °C; $\;{\left[\alpha \right]}_{D}^{25}$ −66 (*c* 0.02, MeOH), UV (MeOH) λ_max_ (log *ε*): 205 (2.51), 260 (1.78) (Appendix A); ECD (MeOH) λ_max_ (∆*ε*): 206 (−18.31), 240 (+12.72), 310 (−2.18); ^1^H (600 MHz, CD_3_OD) and ^13^C NMR (150 MHz, CD_3_OD) data, see Table 1; HR-ESI-MS: *m*/*z* 509.25015 [M+Na]^+^ (calcd. for C_28_H_38_O_7_Na, 509.25097).

Astellolide J (**14**): yellow powder, m.p. 202.9–204.5 °C; $\;{\left[\alpha \right]}_{D}^{25}$ +16 (*c* 0.02, MeOH), UV (MeOH) λ_max_ (log *ε*): 216 (3.15) (Appendix A); ECD (MeOH) λ_max_ (∆*ε*): 203 (−1.82), 225 (+3.27); ^1^H (400 MHz, CD_3_OD) and ^13^C NMR (100 MHz, CD_3_OD) data; HR-ESI-MS: *m*/*z* 305.13565 [M+Na]^+^ (calcd. for C_15_H_22_O_5_Na, 305.13594).

Astellolide Q (**15**): yellow powder, m.p. 160.1–162.2 °C; $\;{\left[\alpha \right]}_{D}^{25}$ +12 (*c* 0.02, MeOH), UV (MeOH) λ_max_ (log *ε*): 218 (3.62) (Appendix A); ECD (MeOH) λ_max_ (∆*ε*): 232 (+5.19); ^1^H (400 MHz, CD_3_OD) and ^13^C NMR (100 MHz, CD_3_OD) data, see Table 2; HR-ESI-MS: *m*/*z* 347.14578 [M+Na]^+^ (calcd. for C_17_H_24_O_6_Na, 347.14578).

### 3.5. ECD Calculation

Firstly, ECD calculations of compounds **1**–**3** and **14**–**15** were performed by the Gaussian 09 program and Spartan’14. Next, the conformations with a Boltzmann population (>5%) were selected for optimization and calculation in methanol at B3LYP/6-31+G (d, p). Finally, the ECD spectra were generated by the program SpecDis 1.6 (University of Würzburg, Würzburg, Germany) and drawn by OriginPro 8.0 (OriginLab, Ltd., Northampton, MA, USA) from dipole-length rotational strengths by applying Gaussian band shapes with sigma = 0.30 eV [29,30].

### 3.6. Bioassays Antimicrobial Activity

Antimicrobial activity assay was performed as previously described in [31,32].

## 4. Conclusions

In summary, three new andrastin-type meroterpenoids (**1**–**3**), one new drimane sesquiterpenoid (**15**) and one sesquiterpenoid J (**14**) that was first isolated from a natural source, together with ten known compounds (**4**–**13**) were isolated from the cultures of the rhizosphere soil of mangrove plant *Avicennia marina* fungus *Penicillium* sp. N-5. Their structures were determined by the analysis of NMR, HR-MS and ECD spectra. All the isolated compounds were investigated for their antimicrobial activities against two phytopathogenic fungi and four bacterial strains. Among them, compounds **1**, **5** and **10** exhibited significant inhibition against *Penicillium italicum* and *Colletrichum gloeosporioides* with MIC values of 6.25, 1.56, 6.25 and 6.25, 3.13, 6.25 μg/mL. Notably, compound **5** showed potential antimicrobial activity against all the strains and the MIC values were lower than 25 μg/mL. Moreover, andrastin-type meroterpenoid antimicrobial activity against phytopathogenic fungi was reported for the first time.

## Figures and Tables

**Figure 1 marinedrugs-20-00514-f001:**
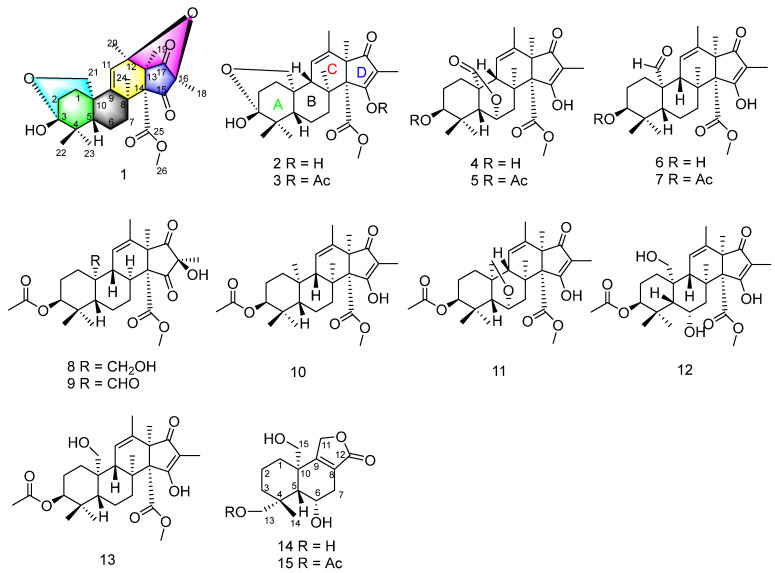
Structure of compounds **1**–**15**.

**Figure 2 marinedrugs-20-00514-f002:**
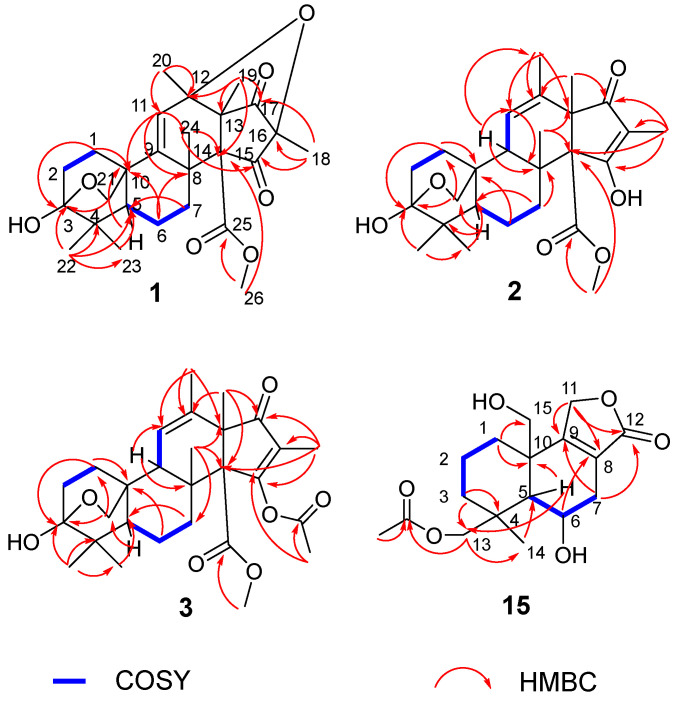
Key HMBC and COSY correlations of **1**–**3** and **15**.

**Figure 3 marinedrugs-20-00514-f003:**
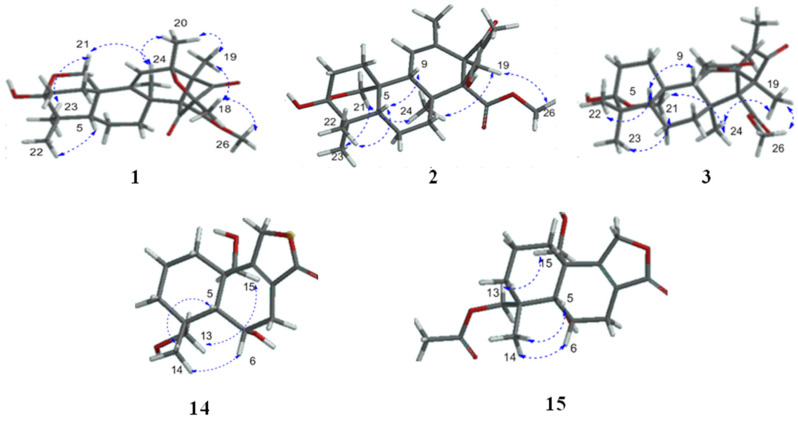
Key NOE correlations of **1**–**3** and **14**–**15**.

**Figure 4 marinedrugs-20-00514-f004:**
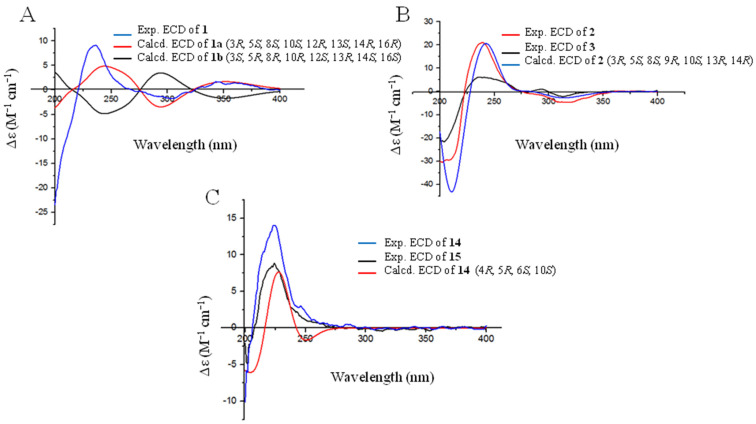
ECD spectra of compounds **1** (**A**), **2** and **3** (**B**), **14** and **15** (**C**) in CH_3_OH.

**Table 1 marinedrugs-20-00514-t001:** ^1^H NMR (600 MHz) and ^13^C NMR (150 MHz) of **1**-**3** in CD_3_OD.

Position	1		2		3	
	*δ* _C_	*δ*_H_ (*J* in Hz)	*δ* _C_	*δ*_H_ (*J* in Hz)	*δ* _C_	*δ*_H_ (*J* in Hz)
1	34.7 (CH_2_)	1.42, m 2.33, m	35.3 (CH_2_)	1.12, m 2.19, m	34.8 (CH_2_)	1.15, m 2.20, m
2	30.2 (CH_2_)	1.85, m 2.16, m	30.3 (CH_2_)	1.74, m 2.12, m	30.1 (CH_2_)	1.76, m 2.14, m
3	99.8 (C)		99.5 (C)		99.5 (C)	
4	41.4 (C)		41.3 (C)		41.3 (C)	
5	49.1 (CH)	1.47, m	50.9 (CH)	1.33, m	51.4 (CH)	1.21, m
6	19.4 (CH_2_)	1.62, m 1.91, m	20.5 (CH_2_)	1.55, m 1.75, m	20.4 (CH_2_)	1.62, m 1.81, m
7	32.3 (CH_2_)	1.94, m 2.61, m	32.5 (CH_2_)	2.07, m 2.76, m	32.9 (CH_2_)	1.94, m 2.23, m
8	40.2 (C)		42.4 (C)		41.9 (C)	
9	150.1 (C)		48.5 (CH)	1.89, t (2.7)	48.6 (CH)	1.98, t (2.7)
10	38.5 (C)		36.3 (C)		36.3 (C)	
11	127.2 (CH)	5.63, s	124.2 (CH)	5.42, m	126.0 (CH)	5.60, m
12	77.4 (C)		137.7 (C)		133.7 (C)	
13	54.4 (C)		57.5 (C)		60.6 (C)	
14	73.4 (C)		69.7 (C)		69.2 (C)	
15	203.7 (C)		190.7 (C)		171.9 (C)	
16	76.7 (C)		113.5 (C)		131.9 (C)	
17	203.8 (C)		201.4 (C)		202.1 (C)	
18	7.9 (CH_3_)	1.19, s	6.6 (CH_3_)	1.57, s	8.8 (CH_3_)	1.55, s
19	11.0 (CH_3_)	1.33, s	18.0 (CH_3_)	1.18, s	17.4 (CH_3_)	1.20, s
20	24.2 (CH_3_)	1.23, s	20.2 (CH_3_)	1.82, s	19.1 (CH_3_)	1.75, s
21	74.4 (CH_2_)	3.55, d (7.6) 4.39, d (8.7)	68.6 (CH_2_)	3.81, d (9.0) 4.22, d (9.0)	68.5 (CH_2_)	3.82, d (8.9) 4.21, d (9.0)
22	27.2 (CH_3_)	1.07, s	27.9 (CH_3_)	1.04, s	27.9 (CH_3_)	1.07, s
23	18.9 (CH_3_)	1.04, s	18.9 (CH_3_)	1.01, s	18.8 (CH_3_)	1.03, s
24	25.9 (CH_3_)	1.49, s	16.7 (CH_3_)	1.19, s	16.5 (CH_3_)	1.24, s
25	169.3 (C)		172.6 (C)		170.9 (C)	
26	52.5 (CH_3_)	3.60, s	51.9 (CH_3_)	3.56,s	52.4 (CH_3_)	3.59,s
Ac-CH_3_					21.2 (CH_3_)	2.36, s
Ac-OCO					167.3 (C)	

**Table 2 marinedrugs-20-00514-t002:** ^1^H NMR (400 MHz) and ^13^C NMR (100 MHz) of **15** in CD_3_OD.

Position	*δ* _C_	*δ*_H_ (*J* in Hz)	Position	*δ* _C_	*δ*_H_ (*J* in Hz)
1	35.3 (CH_2_)	1.45, m 2.05, m	10	44.2 (C)	
2	19.5 (CH_2_)	1.54, m 1.78, m	11	71.8 (CH_2_)	5.00, d (17.7) 4.84, d (17.6)
3	37.0 (CH_2_)	1.16, m 1.91, m	12	177.0 (C)	
4	39.3 (C)		13	68.1 (CH_2_)	4.44, d (11.2) 4.62, d (5.4)
5	56.4 (CH)	1.74, s	14	28.1 (CH_3_)	1.16, s
6	63.6 (CH)	4.61, d (11.0)	15	65.7 (CH_2_)	3.76, d (12.0) 4.33, d (11.9)
7	33.0 (CH_2_)	2.34, d, (18.9) 2.50, d (18.3)	Ac-CH_3_	20.8 (CH_3_)	2.08, s
8	124.0 (C)		Ac-OCO	173.1 (C)	
9	169.0 (C)				

**Table 3 marinedrugs-20-00514-t003:** Antimicrobial activity of compounds **1**–**15**.

	Microbia	Methicillin-Resistent *Staphyococcus aureus* (μg/mL) ^a^	*Bacillus* *subtilis* (μg/mL) ^a^	*Pseudomonas aeruginosa* (μg/mL) ^a^	*Salmonella typhimurium* (μg/mL) ^a^	*Penicillium italicum* (μg/mL) ^a^	*Colletrichum gloeosporioides* (μg/mL) ^a^
Compound							
**1**		25	6.25	>50	>50	6.25	6.25
**2**		>50	>50	25	>50	50	>50
**3**		>50	>50	>50	>50	50	>50
**4**		>50	>50	>50	>50	>50	>50
**5**		50	25	25	>50	1.56	3.13
**6**		>50	25	50	>50	12.50	25
**7**		>50	>50	>50	>50	25	25
**8**		>50	>50	>50	>50	>50	>50
**9**		>50	>50	>50	>50	>50	>50
**10**		25	12.50	25	3.13	6.25	6.25
**11**		>50	>50	>50	>50	>50	>50
**12**		>50	>50	>50	>50	>50	>50
**13**		50	>50	>50	>50	50	>50
**14**		>50	>50	>50	>50	>50	>50
**15**		>50	>50	>50	>50	25	25
Ampicillin		0.13	0.13	0.07	0.13	-	-
Ketoconazole		-	-	-	-	0.78	0.78

^a^: The deviation value of three parallel experiments; -: No test.

## Data Availability

Data are contained within the article and Appendix A.

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
