# Peer review of "Hemiacetalmeroterpenoids A–C and Astellolide Q with Antimicrobial Activity from the Marine-Derived Fungus *Penicillium* sp. N-5"

_marinedrugs, 2022, doi:10.3390/md20080514_

Round 1

Reviewer 1 Report

 1-Comments: Page 2, Line 69: Check the HMBC correlation from H3-20 to the carbonyl C-17.

2- H3-18, H3-19, H3-20, H3-22, H3-23, H3-26, H2-1, H2-2, H2-6, H7-1 instead of H-18, H-19, H-20, H-22, H-23, H-26, H-1, H-2, H-6, H-1. Please check throughout the manuscript

3-Comments: Page 8, Line 204: 75 g instead of 75g; need space between the number and letter. Please check throughout the manuscript

Author Response

Q1: Page 2, Line 69: Check the HMBC correlation from H3-20 to the carbonyl C-17.

Response: Thank you for your suggestion. We have Checked the HMBC correlation from H3-20 to the carbonyl C-17, which is a weak correlation.

Q2: H3-18, H3-19, H3-20, H3-22, H3-23, H3-26, H2-1, H2-2, H2-6, H7-1 instead of H-18, H-19, H-20, H-22, H-23, H-26, H-1, H-2, H-6, H-1. Please check throughout the manuscript.

Response: Thank you for your suggestion. All similar issues have been revised in the manuscript.

Q3: Page 8, Line 204: 75 g instead of 75g; need space between the number and letter. Please check throughout the manuscript.

Response: 75g has been corrected to 75 g. We have checked similar issues in the manuscript in detail.

Reviewer 2 Report

This review concerns the article type manuscript with title: Epoxymeroterpenoids A-C and Astellolide Q with Antimicrobial Activity from the Marine-Derived Fungus Penicillium sp. N-5. The manuscript was submitted to Marine Drugs journal (Manuscript ID: marinedrugs-1871289).

My opinion is valid for structure determination. Four new compounds was isolated from marine fungi and their structures were analyzed mainly using NMR spectroscopy (1H, 13C, DEPT, COSY, NOESY, HMBC, and HSQC) supported by MS spectra. The methodology of structure determination is correct and typical for such study.

In my opinion the manuscript can be accepted for publication, after minor corrections.

My remarks are presented below:

1.       The structures 1-3 cannot be named as epoxy compounds. The name epoxy describes the ring composed of three atoms: 2 carbon atoms and 1 oxygen atom. Moreover, these two carbon atoms cannot be connected with another heteroatom. The structures 1-3 contain structural cyclic motif  -C2-C1-C10-C21-O-C3(OH). Therefore, -O-C3-OH indicates the hemiacetal group, not epoxy group. The structure 1 contains another cyclic ring containing oxygen atom, -C12-C13-C17-C16-O-(C12). Therefore it is 5-membered furane cyclic ether (dihydro-3(2H)-furanone), but not epoxy group. In my opinion, the name of these compounds must be changed. My proposal is hemiacetalmeroterpenoids A-C, but this is the choice of the Authors.

2.       Please, remove spaces in 6, 6, 6, 6, 5, 5-hexa. It should be 6,6,6,6,5,5-hexa

3.       Supporting information. Please, correct Error! Bookmark not defined.

4.       The 1H NMR spectra are not enough clear, in particular in region 1-3 ppm. Please, make zoom of this region in each spectra.

5.       Table 2, position 3. In table is 36.3 in Figure S27 is 36.96, please correct Table 2. Generally, please, check Tables 1 and 2 and gives exactly the same positions of chemical shifts as it is in the spectra.

Author Response

Q1: The structures 1-3 cannot be named as epoxy compounds. The name epoxy describes the ring composed of three atoms: 2 carbon atoms and 1 oxygen atom. Moreover, these two carbon atoms cannot be connected with another heteroatom. The structures 1-3 contain structural cyclic motif  -C2-C1-C10-C21-O-C3(OH). Therefore, -O-C3-OH indicates the hemiacetal group, not epoxy group. The structure 1 contains another cyclic ring containing oxygen atom, -C12-C13-C17-C16-O-(C12). Therefore it is 5-membered furane cyclic ether (dihydro-3(2H)-furanone), but not epoxy group. In my opinion, the name of these compounds must be changed. My proposal is hemiacetalmeroterpenoids A-C, but this is the choice of the Authors.

Response: Thank you for your suggestion. We have renamed compounds 1-3 as hemiacetalmeroterpenoids A-C. All relevant content have also been revised in the manuscript and supplementary materials.

Q2: Please, remove spaces in 6, 6, 6, 6, 5, 5-hexa. It should be 6,6,6,6,5,5-hexa.

Response: This has been corrected as request. 6, 6, 6, 6, 5, 5-hexa was corrected as 6,6,6,6,5,5-hexa.

Q3: Supporting information. Please, correct Error! Bookmark not defined.

Response:  We have corrected the errors in supporting information. Bookmark can be defined now.

Q4:  The 1H NMR spectra are not enough clear, in particular in region 1-3 ppm. Please, make zoom of this region in each spectra.

Response: Thank you for your suggestion. This has been corrected as request in the supporting information.

Q5: Table 2, position 3. In table is 36.3 in Figure S27 is 36.96, please correct Table 2. Generally, please, check Tables 1 and 2 and gives exactly the same positions of chemical shifts as it is in the spectra.

Response: I'm terribly sorry for this problem. δC 36.3 has been corrected to 37.0. We have checked similar issues in the manuscript in detail.